

# PROMICE automatic weather station data

Robert S. Fausto[1], Dirk van As[1], Kenneth D. Mankoff[1], Baptiste Vandecrux[1], Michele Citterio[1], Andreas P. Ahlstrøm[1], Signe B. Andersen[1], William Colgan[1], Nanna B. Karlsson[1], Kristian K. Kjeldsen[1], Niels J. Korsgaard[1], Signe H. Larsen[1], Søren Nielsen[2], Allan Ø. Pedersen[1], Christopher L. Shields[1], Anne M. Solgaard[1], and Jason E. Box[1]

[1]The Geological Survey of Denmark and Greenland, Østervoldgade 10, 1350 København K, Danmark
[2]Deceased

**Correspondence:** Robert S. Fausto (rsf@geus.dk)

**Abstract.** The Programme for Monitoring of the Greenland Ice Sheet (PROMICE) has been measuring climate and ice sheet properties since 2007. Currently the PROMICE automatic weather station network includes 25 instrumented sites in Greenland. Accurate measurements of the surface and near-surface atmospheric conditions in a changing climate is important for reliable present and future assessment of changes to the Greenland ice sheet. Here we present the PROMICE vision, methodology, and each link in the production chain for obtaining and sharing quality-checked data. In this paper we mainly focus on the critical components for calculating the surface energy balance and surface mass balance. A user-contributable dynamic web-based database of known data quality issues is associated with the data products at (https://github.com/GEUS-PROMICE/ PROMICE-AWS-data-issues/). As part of the living data option, the datasets presented and described here are available at DOI: 10.22008/promice/data/aws, https://doi.org/10.22008/promice/data/aws (Fausto and van As, 2019).

## 1 Introduction

The ice loss from the Greenland Ice Sheet has contributed substantially to rising sea levels during the past two decades (Shepherd et al., 2020), and this loss has been driven by changes in surface mass balance (SMB) (Fettweis et al., 2017) as well as in solid ice discharge (Mouginot et al., 2019; Mankoff et al., 2020). SMB changes are typically assessed using regional climate models, but large uncertainties result in substantial model spread (e.g. (Fettweis et al., 2020; Shepherd et al., 2020; Vandecrux et al., 2020)). The spread is especially pronounced in regions of high mass loss (Fettweis et al., 2020). Obtaining in-situ measurements of accumulation and ablation and energy balance in the ablation area are therefore crucial for improving our understanding of surface processes. On-ice automatic weather stations (AWSs) have proven to be the ideal tool to perform such measurements. Presently, the PROMICE AWS data are not included in any reanalysis product such as ERA5, aiding studies with an independent assessment of the performance of regional climate models, and other numerical models that aim to quantify surface mass or energy fluxes (Fettweis et al., 2020).

The Geological Survey of Denmark and Greenland (GEUS) has been monitoring glaciers, ice caps, and the ice sheet in Greenland since the late 1970s (Citterio et al., 2015). Early projects involved ablation stake transects and automated weather



measurements (e.g. (Braithwaite and Olesen, 1989)), however, these efforts could not provide year-round measurements due to accessibility issues and technological limitations. The data these campaigns provided were therefore discontinuous in time and sparse in location. Monitoring programmes using AWSs operating year-round became achievable in the 1990s; the Greenland Climate Network (GC-Net) initiated at Swiss Camp in 1990 and extended to other sites in 1995 (Steffen et al., 1996) and in

1993 AWSs were installed on the K-transect along the southwestern slope of the ice sheet (Smeets et al., 2018). Recently, various institutions installed additional AWSs on the ice sheet, such as at Summit in 2008, and for the Snow Impurity and Glacial Microbe effects on abrupt warming in the Arctic (SIGMA) project in northwest Greenland in 2012 (Aoki et al., 2014). The majority of these AWSs are positioned in the accumulation area of the Greenland ice sheet. The ablation area of the ice sheet was monitored by a handful of stations, underlining the need for a long term monitoring programme for regions of the ice

sheet where melting is the largest mass balance component. Including the PROMICE AWSs in the low-elevation ablation area complements existing monitoring efforts and allows full coverage of the ice sheet, which is necessary to improve understanding of spatiotemporal variability in the surface mass and energy components - key parameters for accurately assessing the state of the ice sheet.

In 2007, the Programme for Monitoring of the Greenland Ice Sheet (PROMICE) was initiated (Ahlstrøm et al., 2008; van As et al., 2011b). GEUS developed rugged AWSs equipped with accurate instruments, and placed them on the Greenland ice sheet and on local glaciers. The AWS design evolved over time with technological advances and lessons learned, but the aim remained to obtain year-round, long-term, and accurate recordings of all variables of primary relevance to the surface mass and energy budgets of the ice sheet surface. The PROMICE monitoring sites were selected to best complement the spatial

distribution of existing ice sheet weather stations, yet within range of heliports and airports.

The development of the PROMICE AWS started at GEUS in 2007 in collaboration with the GlacioBasis Programme monitoring the A.P. Olsen ice cap in northeast Greenland (APO), and the Greenland Analogue Project in southwest Greenland. The AWS is designed to endure extreme temperatures and winds, countless frost cycles, and an ever-changing snow/ice surface, but

with dimensions and weight allowing transportation by helicopter, snowmobile, and dogsled.The original PROMICE network consisted of 14 AWSs, with station pairs in seven regions: Kronprins Christian Land (KPC), Scoresbysund (SCO), Tasiilaq (TAS), Qassimiut (QAS), Nuuk (NUK), Upernavik (UPE) and Thule (THU). Per region the lower (L) station was placed near the ice sheet margin, and the upper (U) station higher up in the ablation area, closer to or at the equilibrium line altitude (ELA) where long-term mass gains and losses are in balance (Figure 1). Other projects collaborating with PROMICE led to the instal-

lation of 11 additional stations (Table 1). Currently some regions also include stations at for instance middle (M) or bedrock (B) sites. Three PROMICE AWSs are located in the accumulation area of the ice sheet (KAN_U, CEN and EGP), whereas two AWSs are on peripheral glaciers (NUK_K and MIT) not connected to the ice sheet. The PROMICE AWSs in Greenland transmit data by satellite in near real-time to support observational, remote-sensing and model studies; weather forecasting; local flight operations; as well as the planning of maintenance visits. The data have been important for quantifying ice sheet

change in e.g. annual international assessment reports such as the Arctic Report Card (Moon et al., 2020b) and the "State



of the climate" (Moon et al., 2020a). The data also proved crucial for calibrating, validating, and interpreting satellite-based observations and regional climate model ouput (Van As et al., 2014; Noël et al., 2018; Huai et al., 2020; Kokhanovsky et al., 2020; Solgaard et al., 2021).

5    The aim of this paper is to describe the PROMICE AWS dataset in detail. We discuss the measurement with insights into post-processing and sensor calibration. The dataset is freely available at www.promice.org (DOI:10.22008/promice/data/aws). We start with a description on how to construct the AWSs, followed by a technical description of the AWS instruments, the data production chain, examples of typical station measurements, and finally a summary and outlook.

## 2    The AWS design

### 10    2.1    The tripod

The AWS tripod is constructed from 32 mm (1.25″) and 44 mm (1.75″) radius aluminium tubes with 3 mm braided stainless steel wires forming a free-standing tetrahedral structure that connects legs and mast in a stable tripod (Figure 2). Most sensors are attached to the 1.7 m long horizontal boom, which is 2.7 m above surface (Figure 2). Weighing c. 50 kg, the battery box, is hanging under the mast to increase the mass of the AWS and to lower its center of gravity for better stability (Table 1). The
tripod can easily be folded to fit in small helicopters. The tripod can also be tilted during maintenance visit for e.g. sensor replacement. Maintenance visits typically take 2-4 hours, which include sensor replacement due for factory recalibration, re-drilling installed sensors in ice, and occasional repairs. Because the tripod is standing freely on the ice surface, it sinks with the melting surface, which results in sonic ranger measurements on the AWS do not capture ice melt. Therefore each PROMICE AWS on ice is accompanied by a separate sonic ranger stake assembly constructed from 32 mm aluminum tubing, typically
drilled 7 m into the ice that does not float with on the ice (Figure 2).

### 2.2    Instrumentation and data transmission

The PROMICE AWS measures 1) the meteorological parameters required for calculating the surface energy budget, 2) snow and ice ablation/accumulation, 3) sub-surface temperature at 8 depths (Thermistor string, Figure 2), and 4) position by GPS. The next section provides details on the frequency and accuracy of measurements taken by each sensor. Further sensor details
are provided in the Appendix.

Measurements are taken every 10 minutes and stored in the data logger locally. The AWSs transmit hourly averages based on 10-minute measurements during the period with ample solar power, between day-of-year 100 and 300 (10 April and 26 October in non-leap years). Exceptions are parameters with low variability (GPS position, station tilt, surface height, etc.) that
are transmitted less frequently (every 6 hours) in order to reduce transmission cost. In winter, between day-of-year 300 and 100, the stations only transmit daily averages of all parameters to limit power consumption by the satellite modem. Trans-

mission is done through the Iridium satellite network that has coverage even at the northernmost latitudes. The Iridium short burst service transmits up to 340 bytes per message. The program running on the data logger ensures a correctly transferred data string from the logger to the transmitter if an Iridium satellite is in view. In case of no successful transmission through the satellite the logger program will try again. Depending on the availability of the Iridium service the logger program can

also queue the message for delivery at a later time with better satellite connection. This relatively low power operation mode ensures unnecessary transmission attempts with a low rate of message loss. Moreover, the logger program encode the data in a binary format before transmission, which reduces the size of the message and by that transmission costs with about 2/3.

To ensure reliable and accurate measurements, instruments in the field are swapped following an instrument maintenance

schedule based on information from manufacturers and from experience, for instance with battery life and performance, when charging batteries without a charge regulator. The maintenance schedule is a guideline; not always does a field crew return to an AWS in time for a scheduled sensor swap. For example, we only visit the AWSs in the north-eastern part of Greenland (KPC, 1) every 3-4 years as their remoteness weighs heavily on the logistics budget. Thankfully, the most remote PROMICE AWSs experience less melt, lower accumulation and weaker storms than some other places, reducing the need for maintenance

visits.

## 3    Measurements

### 3.1    Dataset pruduction chain

PROMICE AWS data are processed by the production chain algorithm with some manual expert quality checking twice a year (typically in January and after the summer), and in real time with automated quality check in the PROMICE database. For our

production chain algorithm, we make use of the raw data recorded every 10 minutes, retrieved from the data logger during maintenance visits (Figure 3). For the period since the last station visit we use the transmitted data for the PROMICE data products. Beside the direct AWS measurements, we also calculate certain variables based on these measurements, for instance tilt-corrected solar radiation and turbulent heat fluxes. In the following, we describe each variable in the PROMICE AWS dataset, and how it is measured or derived. We refer to the manufacturer specific instrument information, accuracy, and power

consumption (see Table 2 and sensor specific tables in Appendix). We use simple thresholds on 10-min data to remove spikes and inconsistent or bad measurements (see section "Post processing" below for more information). Available transmitted data are used for filling out data gaps.

### 3.2    Measured variables: description and uncertainty

For most measured variables, the data logger converts readings in voltage to physical values using simple scaling relations with

calibration coefficients specific for each instrument. Only when identical sensors can have different calibration coefficients, namely the radiometer and pressure transducer, the conversion from voltage is done in post-processing; the advantage being

that a sensor swap does not require a data logger program change in the field. Below, we mention all scaling relations needed to manually convert logger data to physical measurements.

**Air pressure**

Barometric pressure (unit: hPa) is measured in the fiberglass-reinforced polyester logger enclosure (Figure 2, number 9). The
logger enclosure is generally located 1.5 m above the ice surface. The barometer manufacturer reports a measurement accuracy of ±2 hPa within the -40 to +60 °C temperature range (Table 2, see also Appendix for more information).

**Air temperature**

Air temperature (unit: °C) is measured inside a fan-aspirated radiation shield (Figure 2, number 6). The sensor is located approximately 2.6 m above the ice surface, i.e. as high as possible underneath the sensor boom. The measurement height
varies when a winter snow cover is present. The temperature sensor is a PT100 probe that changes its electrical resistance with temperature and has an accuracy of ±0.1 °C (Table 2, see also Appendix for more information). Also a secondary air temperature reading (°C) is made in the aspirated shield from the Hygroclip temperature/humidity sensor described in the following, also with a manufacturer stated accuracy of ±0.1 °C, but we consider the Hygroclip temperature to be less accurate than the PT100, given the need for more frequent sensor re-calibrations.

**Humidity**

Relative humidity (RH) (unit: %) is measured alongside the PT100 in the aspirated radiation shield using a HC2A-S3 (or HC2) "Hygroclip" (Figure 2, number 6). The sensor measures relative humidity with ±0.8 % accuracy. Relative humidity is measured relative to water. For temperatures below freezing, relative humidity is recalculated relative to ice in post-processing (see section 3.3). To distinguish between the two relative humidities in the PROMICE data products, the prior humidity (unadjusted below
freezing) is called "relative humidity with respect to water", while the latter simply is "relative humidity". The conversion of relative humidity relative to ice is after Goff and Gratch (1946). Every 1-2 years, the Hygroclip is replaced by a sensor re-calibrated in a closed chamber at room temperature with constant relative humidities of 10, 35, 80 %.

**Wind speed and direction**

Wind speed and direction (Units: m s$^{-1}$ and °) measurement height is approximately 3.1 m above the ice surface, and like the
other measurements has a reduced measurement height in case a winter snow layer is present (Figure 2, number 4). An AC sine wave voltage signal is produced by the rotation of the four-bladed propeller, and the pulse count converts to wind speed using a multiplier. According to the manufacturer the sensor can measure wind speeds between 0 and 100 m s$^{-1}$, with an accuracy of ±0.3 m s$^{-1}$ or 1% if the measured value is higher than 30 m s$^{-1}$.

Wind direction is measured through changes in the vane angle by a precision potentiometer housed in a sealed chamber on the





instrument. The output voltage is directly proportional to vane angle wind direction is measured between 0 - 360° with an accuracy of ±3°. Every three years the sensor is replaced and tested for drift and functionality with an "anemometer drive" rotating the propeller at a known rate. The instrument's orientation is logged and reset to "Geographic north " during each maintenance visit to keep wind direction data accurate within ±15° (although much larger station rotations have been encountered).

## 5 Upward and downward shortwave radiation

Horizontally-leveled up and down facing Kipp & Zonen CNR1 or CNR4 record solar radiation (units: W m$^{-2}$), respectively. Measurement height is at the sensor boom level of 2.7 m over the ice surface (Figure 2, number 1). Shortwave radiation is measured by the pyranometers within plastic meniscus domes, allowing minimal water droplet adhesion. The manufacturer reports that sensor uncertainty is 10%. This sensor uncertainty has in practice been found to be ca. 5% for daily totals in
Antarctica ((van den Broeke et al., 2004)). The radiometers are recalibrated at Kipp & Zonen every three years. The radiometer is one of the few variables stored in the data logger in voltage units, because every radiometer has a different set of calibration coefficients, whereas all logger programs running on PROMICE AWSs are identical, for practical reasons. In post-processing, sensor readings SR_raw are converted into a physical measurement SR_m following:

$$SR_m = \frac{SR_{raw}}{C_{SR}}, \tag{1}$$

where $C_{SR}\ V(Wm^{-2})^{-1}$ is a sensor calibration coefficient and SR_m is either the converted downward and upward shortwave irradiance. Shortwave radiation measurements are corrected for sensor tilt following van As et al. (2011a) in post-processing, which means that the PROMICE AWS dataset contains both uncorrected and corrected values.

### Upward and downward longwave radiation

Longwave radiation (units: W m$^{-2}$) is also measured by the CNR1/CNR4 radiometer mounted at approximately 2.7 m over
20 the ice surface (Figure 2, number 1). The radiometer contains a pair of up- and downfacing pyrgeometers, with a spectral range of 4.5 to 42 μm. As for shortwave radiation, longwave radiation is stored in voltage units (LR_raw) in the data logger, and transformed to physical units (LR_m) in post-processing following:

$$LR_m = \frac{LR_{raw}}{C_{LR}} + 5.67 \cdot 10^8 \cdot (T_{rad} + T_0)^4, \tag{2}$$

where $C_{LR}\ V(Wm^{-2})^{-1}$ is the sensor calibration coefficient. $T_{rad}$ is the sensor temperature measured in the radiometer
casing in °C and $T_0 = 273.15$ °C.

### Surface height

The height of the sensor boom (units: m) is measured by a sonic ranger attached to the boom itself attached approximately 0.1 m below the boom (2, number 5a), while the height of the stake assembly is measured about 0.1 m below an aluminum boom connecting stakes drilled into ice (2, number 5b). The sensor outputs a distance (H_raw) that requires an air temperature





correction in post-processing. The temperature adjustment is performed following:

$$H_m = H_{raw} \cdot \sqrt{\frac{T_{air} + T_0}{T_0}} \tag{3}$$

After temperature correction, the measurement uncertainty of the SR50A sonic ranger reported by the manufacturer (Campbell Scientific) is ± 1 cm or ± 0.4% of the measured distance. The uncertainty of sonic ranger readings in PROMICE was investigated utilizing data from a wintertime accumulation-free period of more than 2 month at the location SCO_U. The associated standard deviations for the two sensors were found to be 1.7 cm and 0.6 cm after spike removal, amounting to 0.7 % and 0.6 % of the measured distance, respectively ((Fausto et al., 2012)). In addition to the sensor uncertainties, occasional problems with the stake assembly occurred, primarily in terms of stability during storms when melted out several meters. Also, an unknown amount of melt-in of the stake assembly can occur, but we speculate this only happens 1) when surface melt since installation has been considerable, increasing the height of and thus pressure applied by the stake assembly, and 2) when the stake bottoms are not plugged with caps, as was only the case until 2010.

The PROMICE AWSs are also equipped with a pressure transducer assembly (PTA) that measures surface height change due to ice ablation (2, number 7). The assembly was first constructed and implemented in Greenland in 2001 by Bøggild et al. (2004), but was further developed within PROMICE ((Fausto et al., 2012)). The PTA consists of an antifreeze-mixture-filled hose with a pressure transducer attached at the bottom. Drilling the hose typically more than 10 m into the ice, the pressure signal registered by the transducer will be that of the vertical liquid column over the sensor, where the upper level is a bladder fixed on the tripod in a shielded box. This allows in-/outflow antifreeze due to compression while keeping a steady level at roughly 1.5 m above the ice surface depending on the AWS. 2 illustrates the free-standing AWS tripod that floats on the ice surface and moves down with the ablating surface, while the hose itself melts out of the ice, which in turn will reduce the hydrostatic pressure from the vertical liquid column over the pressure transducer at the bottom of the hose. The measured reduction in pressure at the bottom of the hose translates directly into ice ablation. As for the radiometer, every pressure transducer has a different calibration coefficient, which is why measurements are stored in the data logger in voltage units and transformed to a physical measurement in post-processing. Measurement height ($H_m$), or in fact depth relative to the PTA bladder, is calculated as follows:

$$H_m = C_{PTA} \cdot \frac{\rho_w}{\rho_{af}} \cdot H_{raw}, \tag{4}$$

where $C_{PTA}$ is the calibration coefficient. The constants $\rho_w$ and $\rho_{af}$ are the densities of water and the 50/50 antifreeze solution, respectively.

**Sub-surface temperature**

Subsurface temperatures (unit: °C) are measured by a 10 m thermistor (temperature-dependent resistor) string (Figure 2, number 11). The string measures at 1, 2, 3, 4, 5, 6, 7 and 10 m depth, although depths vary due to the surface ablation and accumulation. The string is constructed at GEUS (see Appendix for more information).





**Station tilt**

The inclinometer is installed on the sensor boom (Figure 2, number 2) and is aligned with the radiometer to allow for tilt correction of shortwave radiation measurements. The inclinometer measures the tilt (unit: °) across (left-right) and along (up-down) the sensor boom, which translates into tilt-to-east and tilt-to-north when the sensor boom is perfectly oriented
north-south. The tilt sensor readings in voltage units ($Tilt_{raw}$) are converted into tilt in degrees following:

$$Tilt_m = 21.1 \cdot |Tilt_{raw}| - 10.4 \cdot |Tilt_{raw}|^2 + 3.6 \cdot |Tilt_{raw}|^3 - 0.49 \cdot |Tilt_{raw}|^4, \tag{5}$$

where all constants were determined at GEUS (Table 2). Ice ablation causes the AWS tripod to melt downward; this changing (slippery) surface often results in AWS tilt changes of more than several degrees.

**AWS position**

We use a single frequency GPS receiver to measure the position (units: °N/°W) and the elevation (unit: m above sea level) of each station to quantify ice dynamics (Figure 2, number 9). The GPS antenna, as well as the receiver which is contained in the Iridium modem, is placed inside the data logger enclosure. It is a single frequency GPS, which is built into Iridium 9602-LP modem. The receiver type is a NEO-6Q, 1575.42 MHz (L1), 16-channel, with a C/A code. The accuracy is reported to be within 2.5 m. In the PROMICE AWS setup, the GPS receiver is powered up for 5 minutes preceding each Iridium transmission
(hourly in summer and daily in winter), during which it attempts to acquire location data every 20 seconds. The return (out of a maximum of 15) that reports the lowest horizontal dilution of precision is written to memory. So far, NUK_U, NUK_L, MIT and QAS_L have been repositioned during maintenance visits over distances larger than several tens of meters. The main reason for this is to reduce the influence of location change on the AWS variables measured, but stations have also been relocated to move them away from a region with opening crevasses. Table 3 shows the horizontal and vertical displacement
due to glacier flow and AWS relocation during maintenance visits.

### 3.3  Post processing

In this section, we describe and quantify the filtering process, how we correct measurements, and how we calculate derived variables in the dataset. The hourly, daily, and monthly averaging procedures are also described.

### 3.3.1  Filtering

Table 4 provides filtering information used in the processing chain. We remove unrealistic spikes from the data by using upper and lower thresholds for each measurement. Measurements outside these (generous) threshold limits, which could happen for a number of known and unknown reasons, are considered erroneous and set to -999. Known reasons will be discussed in section 4.2 (Living data section). Derived variables are also set to -999 when one or more of the listed "core" AWS measurements that serve as input fall outside the threshold limits.



### 3.3.2 Derived and corrected variables

**Specific humidity**

The specific humidity $q$ (unit: kg/kg) is calculated from relative humidity with respect to water/ice above/below freezing (RH) using the following equation:

$$q = \frac{RH}{100} \cdot q_{sat}, \tag{6}$$

with

$$q_{sat} = \frac{\epsilon \cdot es_{ice/water}}{p - (1 - \epsilon) \cdot es_{ice/water}}, \tag{7}$$

where $\epsilon = 0.622$ is the ratio between the specific gas constants for dry air and water vapor, $p$ is air pressure in Pa and $es_{ice/water}$ is saturation water vapor pressure over ice in Pa (below freezing) or water (above freezing) calculated after Goff and Gratch (1946).

**Surface temperature**

The surface temperature $T_s$ (unit:°C) is derived using the measured downward and upward longwave irradiance ($LR_{in}$ and $LR_{out}$, repetively):

$$T_s = \left( \frac{LR_{out} - (1 - \epsilon) \cdot LR_{in}}{\epsilon \cdot 5.67 \cdot 10^{-8}} \right)^{0.25} - 273.15, \tag{8}$$

where ice sheet surface emissivity $\epsilon = 0.97$.

**Turbulent energy fluxes**

The sensible (SHF) and latent (LHF) heat fluxes (unit: W m$^{-2}$) are estimated using vertical gradients in wind speed, potential temperature, and specific humidity between the measured boom height and the surface described by Van As et al. (2005); Van As (2011). According to the Monin-Obukhov similarity theory, SHF and LHF can be approximated as:

$$SHF = \rho C_p \kappa^2 \frac{u}{ln\frac{z_u}{z_0} - \psi_u} \frac{T - T_s}{ln\frac{z_T}{z_{0,T}} - \psi_T}, \tag{9}$$

$$LHF = \rho L_{s/v} \kappa^2 \frac{u}{ln\frac{z_u}{z_0} - \psi_u} \frac{q - q_s}{ln\frac{z_q}{z_{0,q}} - \psi_q}, \tag{10}$$

Here $\rho$ is the density of air and $C_p = 1005\ JK^{-1}kg^{-1}$ its specific heat capacity at constant pressure. $L_s = 2.83 10^{-6}\ Jkg^{-1}$ and $L_v = 2.50 10^{-6}\ Jkg^{-1}$ are the latent heat values of sublimation and evaporation, respectively, while $\kappa = 0.4$ is the von Karman constant. When estimating turbulent heat fluxes, we need the measurement heights ($z_u$, $z_T$, $z_q$, Table 2) of wind speed (u), temperature (T), and specific humidity (q) as well as the surface roughness lengths for momentum $z_0$, for heat, $z_{0,T}$ and moisture $z_{0,q}$. We use $z_0 = 0.001$ m and $z_{0,T} = z_{0,q}$ are calculated using the formulation from Smeets and Van den Broeke (2008a, b) for rough surfaces. The stability correction functions $\psi_{u,T,q}$ Eq. 12 in Holtslag and De Bruin (1988) for stable atmospheric conditions, while we follow Paulson (1970) for unstable conditions. The surface temperature ($T_s$) is calculated





from longwave radiation (see Eq. (8)) and the surface specific humidity is assumed to be at saturation ($q_s = q_{sat}$).

Several sources of uncertainty apply to the calculation of SHF and LHF. The aerodynamic surface roughness length $z_0$ is known to vary with surface type (Brock et al., 2006) and through time (Smeets and Van den Broeke, 2008a, b). Using the constant value of $z_0 = 0.001$ m could be an overestimation of surface roughness in presence of snow and subsequently lead to an overestimation of both turbulent fluxes. Since most PROMICE stations are located in the ablation area, the snowpack is melted during spring and the surface becomes snow-free for most of the ablation season. The calculation of surface temperature also relies on certain assumptions (see section above). Several studies evaluated the performance of the Monin-Obukhov similarity theory in Greenland. Using two and single profile level methods versus eddy covariance and evaporation lysimeters, Box and Steffen (2001) found underestimation of downward LHF during extreme stability cases. Miller et al. (2017) used a similar method for calculating SHF and reported a root mean square difference (RMSD) of $8.7\ Wm^{-2}$, with average bias of $-7.0\ Wm^{-2}$, when compared with their two-level Eddy Covariance estimation of SHF. Miller et al. (2017) emphasize that SHF records by one-level approaches often cover longer time periods. Fausto et al. (2016b, a) found they had to use an unrealistically high $z_0$ to get agreement between SEB closure and observed ablation rates during extreme sensible and latent heat-driven melt events.

**Tilt correction of downward shortwave radiation and Cloud Cover**

Tilt correction of solar radiation is performed following Van As (2011). Downward shortwave radiation ($SR_{in}$) consists of a diffuse and direct beam part. It is only the direct beam part of $SR_{in}$ that requires tilt correction. We have for a horizontal radiation sensor that the direct beam equals $SR_{in}$ reduced by its diffuse fraction ($f_{dif}$). For the tilted radiation sensor, $SR_{in}$ is calculated from the measured value, $SR_{in,m}$ and a correction factor C following:

$$SR_{in,cor} = SR_{in,m} \frac{C}{1 - f_{dif} + Cf_{dif}}, \tag{11}$$

with

$$C = \cos(SZA) \cdot \Big( \sin(d)\sin(lat)\cos(\phi_{sensor}) - \sin(d)\cos(lat)\sin(_{sensor})\cos(\phi_{sensor}) +$$

$$\cos(d)\cos(lat)\cos(\theta_{sensor})\cos(w) +$$

$$\cos(d)\sin(lat)\sin(\theta_{sensor})\cos(\phi_{sensor})\cos(w) + \cos(d)\sin((\theta_{sensor})\sin(\phi_{sensor})\sin(w) \Big)^{-1} \tag{12}$$

where $SZA$ is the solar zenith angle, $d$ is the sun declination (the angle of the sun above the plane formed by the earth's equator), $w$ is the hour angle (the angle between the sun's current position in the sky and its position at solar noon), $lat$ and $lon$ are the site's latitude and longitude in radians, and lastly $\theta_{sensor}$ and $\phi_{sensor}$ are the radiometer's tilt angle and direction, respectively. The calculation procedures for $d$ and $w$ and $SZA$ are detailed in NOAA (2020). We estimate $f_{dif}$ spanning from 0.2 for clear skies to 1 for overcast conditions, while assuming a linear dependency on the cloud cover fraction (Harrison





et al., 2008). We approximate the cloud cover fraction from its dependence of the near-surface air temperature ($T_{air}$) on $LR_{in}$ (Van As et al., 2005). For this purpose, we calculate a theoretical downward longwave radiation flux corresponding to clear sky conditions using the equation from Swinbank (1963):

$$LR_{clear} = 5.31 \cdot 10^{-14} \cdot (T_{air} + T_0)^6, \tag{13}$$

and to overcast conditions assuming the black-body radiation:

$$LR_{overcast} = 5.67 \cdot 10^{-8} \cdot (T_{air} + T_0)^4. \tag{14}$$

The cloud cover (limited to the [0:1] range) is then calculated as:

$$cloudcov = \frac{LR_{in} - LR_{clear}}{LR_{overcast} - LR_{clear}} = \frac{f_{dif} - 0.2}{0.8} \tag{15}$$

**Albedo**

Surface broadband solar reflectivity in the 0.3 to 2.5 $\mu$m wavelength range, a.k.a., albedo (unitless) is calculated from 10 minute tilt-corrected downward and upward solar irradiance data. Hourly averaged albedo values are calculated for cases when the sun hits the radiometer top in angles exceeding 20, i.e. when measurements are most reliable for this sensor type. Daily albedo averages are computed from available hourly data. AWS obstruction of sunlight, casting a shadow within the radiometer's field of view, may lower the albedo on average by 0.03 (Kokhanovsky et al., 2020), but depends on surface type and height. Also

of relevance to measured albedo are the contrast of the surface relative to the AWS battery box, legs, mast and enclosure, and whether a melt pond forms beneath the AWS Ryan et al. (2017) examined spatial variograms in UAV-derived albedo versus satellite and PROMICE albedo and found differences increasing for some PROMICE sites toward the late melt season when the AWS point measurements lack representativity of the increasingly inhomogeneous surface cover. van den Broeke et al. (2004) found a 5% uncertainty on pyranometer measurements, while the manufactorer, Kipp & Zonen, estimate a more conservative

value of 10% uncertainty. We conservatively assume 10 % uncertainty in the calculated albedo.

**Ice surface height**

The pressure transducer assembly (PTA, Fig 2, sensor 7) setup is influenced by variations in air pressure. The air pressure contributions to the measured PTA signal $H_M$ are eliminated using the following equation:

$$H_L = H_M + \frac{P_C - P_A}{g\rho_l}, \tag{16}$$

where $P_A$ (Unit: hPa) is air pressure, $P_C$ (Unit: hPa) is the known pressure given by the manufacturer to which the sensor was calibrated, $g = 9.82$ m s$^{-2}$ is the gravitational acceleration, and $\rho_l = 1090$ kg m$^{-3}$ is the antifreeze mixture density at 0 °C. Changes in $H_L$ are equal to ice ablation. Fausto et al. (2012, 2016a) compared PTA time series to hose measurements manually performed in field and recorded distances from sonic rangers to quantify instrument inaccuracies, which were found to be accurate within 0.04 m.

### 3.3.3 Averaging

The time reported in our data products specifies the hour/day/month during which the measurements are taken, as opposed to other products that list the exact timestamp of the end of the averaging period. Hourly averages are calculated from 10-min values if at least one value is available (10-minute data are seldomly missing). We then calculate daily averages from hourly

averages if at least 20 values (∼80%) are available for a dataset variables with a clear diurnal variability. Less transient variables require at least one measurement to calculate an average. Lastly, we calculate the monthly averages from daily averages if at least 24 values (∼80%) are available.

### 3.3.4 Measurement success rate

To illustrate the PROMICE AWS data coverage, we determined the "success rate" in terms of available daily averages for

all measured variables that are required for estimating the surface energy budget: air pressure, air temperature, humidity, wind speed, and down- and upward short- and longwave radiation. Success rate is defined as the ratio of the counts of not-flagged and the number of days since AWS installation. The performance for the critical variables for each station and their measurement periods is illustrated in Figure 4. 18 out 26 stations have at least a 85% success rate for all critical surface energy budget variables, while six have experienced significant periods with power failure, station toppling, snow accumulation exceeding

instrument height, or even crevasse formation underneath the station.

## 4 Data products and availability

The PROMICE AWS data are made available in hourly (H), daily (D), and monthly (M) time resolutions. The data products include variables listed in Table 5. The data are organised in ASCII files, organized following Table 5, with 46 columns in the hourly datafiles, 45 columns in the daily datafiles, and 24 columns in the monthly datafiles. The datafiles can be accessed

through "Download Data" on the PROMICE webpage https://www.promice.org (DOI: 10.22008/promice/data/aws).

### 4.1 Data examples

To create a quick insight into the data product, we show examples of data from AWSs in two contrasting locations: TAS in southeast Greenland near Tasiilaq and UPE in northwest Greenland near Upernavik (Figure 1).

### 4.1.1 Wind speed

Time series spanning the years 2012 through 2014 of weekly median wind-speeds and maximum 10-min wind speed within that week for TAS_L and UPE_U are displayed in Figure 5. Median wind speeds are lower at TAS_L than at UPE_U, whereas the opposite is true for maximum wind speeds, because TAS_L is located in a region well-known for its Piteraq storms.

### 4.1.2 Air temperature

The daily average air temperature for the two stations near Upernavik is shown in Figure 6. The temperature is higher at the lower station, UPE_L, than at the upper station, UPE_U due to an elevation difference of more than 700 m. The tendency of the temperature to have a higher variability during winter months than during summer is also evident from these time series.

### 4.1.3 Surface energy balance

Figure 7 presents the surface energy fluxes at UPE_U in 2012. The plots show how UPE_U experiences a shorter period with solar radiation, due to the more northerly location, when comparing to the TAS stations further south. Furthermore, Figure 7 shows how the outgoing longwave radiation becomes stable during the main melt season when the surface temperature is at the melting point. The sum of all the fluxes determines if there is a surplus of energy at the surface, which can be used for snow or ice melt.

### 4.2 Living data and continuing improvements

PROMICE will continue to update and make available the data products as AWS data comes in. It is likely that there are as-yet-unknown issues in the existing data we are releasing as part of this dataset, and new issues may arise in as-yet-to-be collected data. Also, some issues are known but hard to identify and some issues are systematic, which can be corrected for more generally. Below, we list known dataset issues in three categories: 1) Issues that are hard to identify 2) issues we in some way can correct for systematically, and 3) Errors caused by humans, animals, and anything due to instrument failure.

Here we list dataset issues we have encountered over the years following the above three categories:

1. Hard to identify

    (a) Often a high inclinometer variability, presumably caused by AWS shaking, or instrument failure.

    (b) Riming affecting several measured variables.

    (c) Undocumented AWS orientational drift.

    (d) Sonic ranger membrane not robust enough to always survive the period between maintenance visits (instrument failure).

    (e) Instruments buried in snow during winter and/or spring.

    (f) Tripod collapse due to compacting snow.

    (g) AWS falling over in extreme winds or crevassed terrain.

    (h) Bent sensor boom due to compacting snow, impacting alignment radiometer and inclinometer.

    (i) Leaks in or overfilling of the pressure transducer assembly.



    (j) Static electricity by snow drift or damage to the AWS's electrical circuit.

2. Systematic correction

    (a) Radiometer sensor tilt (we already correct for this).

    (b) Glacial movement causing gradual changes in AWS positions and thereby their measured variables

    (c) Shading by instruments and station frame impacting measurements, e.g. albedo.

3. Errors caused by humans and animals

    (a) Human error, such as sensor plug swap during maintenance visits or improper (wrong height/orientation) sensor mounting.

    (b) Animal occasionally soiling instruments and AWS surroundings.

    (c) Various instrument failures.

The most recent data files will in most cases comprise of transmitted data, which will be updated after the next maintenance visit. Data download from the logger will improve data quality and coverage. During strong winds the AWSs can topple or sensors break down. AWSs can also be covered by winter-accumulated snow, in which cases will reduce the data quality for many variables. We can with our height measurements on both the station and stake assembly monitor when certain instruments are covered in snow. At present, AWS covered in snow has only happened at three locations, namely QAS_U, QAS_M and MIT. Data recorded after and during these events are often identified by the automatic processing routine and will be clearly identifiable for the data user as erroneous data. A maintenance visit either in spring or summer will often result in a station being moved, leveled, and/or rotated, in which case variables such as surface height will undergo an easily recognizable shift.

Identified dataset issues that we plan to correct for or implement in future data products:

1. Shading by instruments and station frame impacting albedo.

2. Instrumental monitoring of AWS orientation, which could influence the correction of the shortwave radiation and wind direction.

3. Instrumental monitoring of rain.

4. Flagging protocol for identified errors and issues.

While we do our best to clean the data appropriately and address known issues (see above), we recognize that correcting issues is more complicated than simply documenting them, and that some corrections may not be possible, or may be subjective and a function of different use cases. We therefore introduce a user-contributable dynamic web-based database of known data quality issues at (https://github.com/GEUS-PROMICE/PROMICE-AWS-data-issues/). The current implementation uses GitHub "issues", although a future version may use a different database backend that the DOI would resolve. Each issue is





tagged with station(s), sensor(s), and year(s) where the issue occurs. Users who are working with a station, sensor, or time-frame of data are encouraged to search the issue database and see if there are any known relevant data issues. If users discover a data issue that is not currently documented, they can add it to the database. A PROMICE team-member will review and tag any issues as verified, and then suggest a fix. Future versions of the product will implement these fixes if possible, and the
issues will be closed but remain accessible.

## 5  Summary and outlook

The UN Inter-Governmental Panel on Climate Change (IPCC) has previously highlighted the value of station-level records for assessing the cryospheric changes associated with global climate change (Vaughan et al., 2013). The IPCC has more recently highlighted the importance of understanding Greenland ice sheet mass loss, especially mass loss due to atmospheric forcing
and surface mass balance mechanisms, as a leading contributor to sea-level rise (Meredith et al., 2019). Meteorological and glaciological monitoring sites on the ice sheet are necessary to provide well-constrained observations of surface energy and mass balances. Understanding these local energy and mass balances provides the process-level knowledge of ice sheet and atmosphere interactions required by regional and global simulations e.g. (Van As, 2011; Fausto et al., 2016b). The PROMICE network plays a leading role in providing these in situ observations and process-level insights for the Greenland ice sheet.

The PROMICE AWS v3 data products are made available as hourly, daily, and monthly datafiles. All data products undergo periodical improvement through updates in the processing chain. Data are added as they are received from field parties and through satellite transmission. Between 2007 and 2021, the PROMICE AWSs have measured with a success rate of 85% for 18 out 26 stations, defined as the availability fraction of the daily averages for variables required for calculating the surface energy balance (see Figure 4). All PROMICE AWS data products are available at https://doi.org/10.22008/promice/data/aws.

In addition to advancing science, the PROMICE AWS network is now poised to contribute to operational products. With recent advances in the quality and transparency of the PROMICE data delivery pipeline described here, as well as the increasing prevalence of machine-to-machine transfer protocols among data users, the entire PROMICE station data archive – including near real-time observations – is now readily available to ingest in weather forecast and climate reanalysis applications. With the original AWS stations quickly approaching its fifteenth anniversary, the PROMICE data record is crossing the halfway
mark of a thirty-year climatological reference period. With the launch of the PROMICE AWS data issues on GitHub (https://github.com/GEUS-PROMICE/PROMICE-AWS-data-issues/), we hope to continue to support the growing PROMICE user community into the next decade.

## 6  Data availability

The PROMICE AWS product has DOI: https://doi.org/10.22008/promice/data/aws. Fausto and van As (2019) is the dataset
citation.



*Author contributions.* RSF and DVA produced the PROMICE AWS product. RSF and KDM set up the data-curation framework. RSF prepared the manuscript with contributions from all co-authors, except SN who is deceased.

*Competing interests.* Authors declare that they have no conflict of interest.

*Acknowledgements.* AWS data from the Programme for Monitoring of the Greenland Ice Sheet (PROMICE) and the Greenland Analogue
5    Project (GAP) were provided by the Geological Survey of Denmark and Greenland (GEUS) at https://www.promice.org.





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

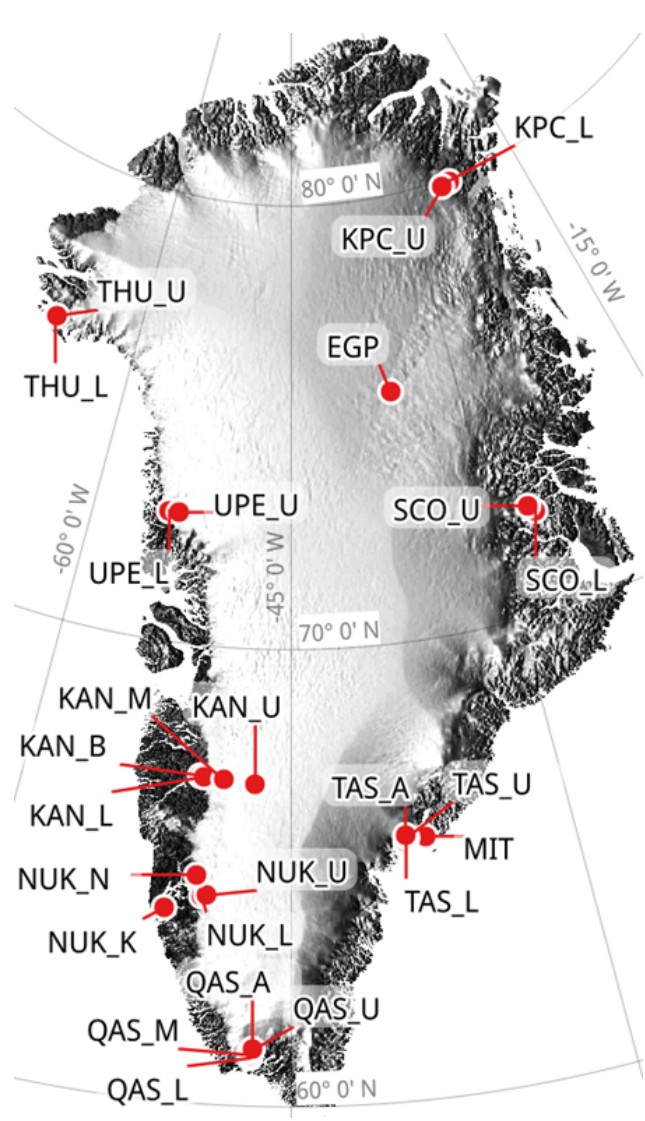

**Figure 1.** Map of Greenland with PROMICE automatic weather station locations.

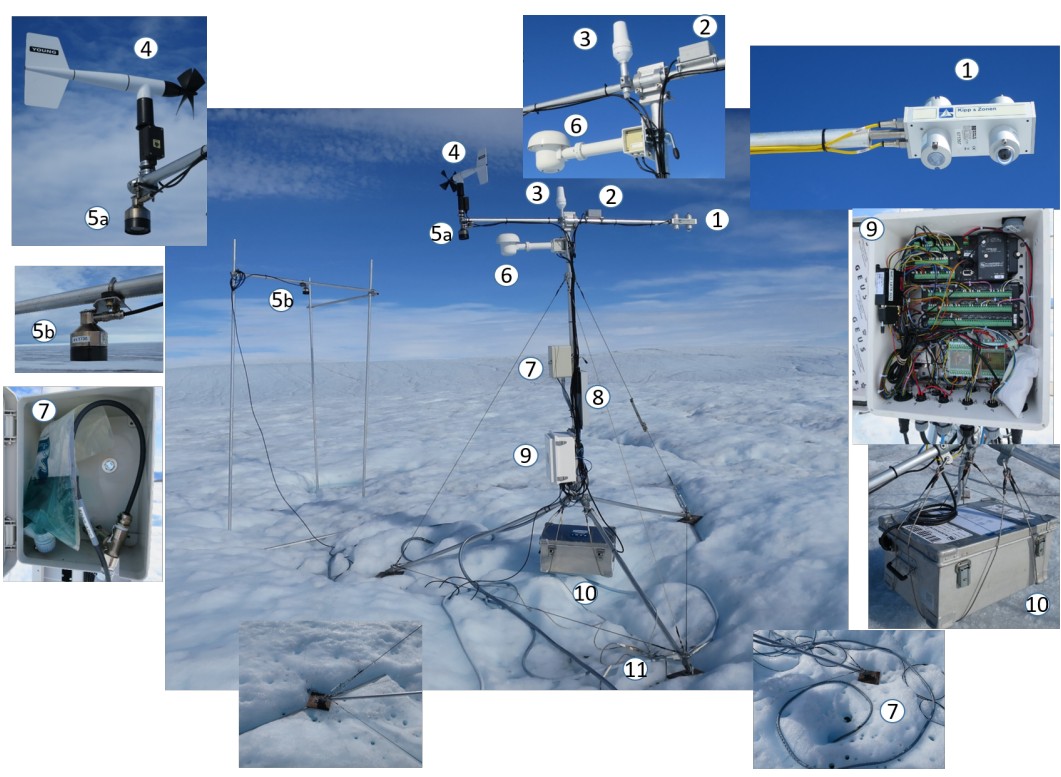

**Figure 2.** PROMICE automatic weather station UPE_U photographed on 4 August 2018. 1: Radiometer, 2: Inclinometer, 3: Satellite antenna, 4: Anemometer, 5: Sonic rangers, 6: Hygro-/thermometer (aspirated), 7: Pressure transducer, 8: Solar panel, 9: Data logger, multiplexer, barometer, satellite modem and GPS antenna, 10: Battery box, 11: Thermistor string (8 levels).





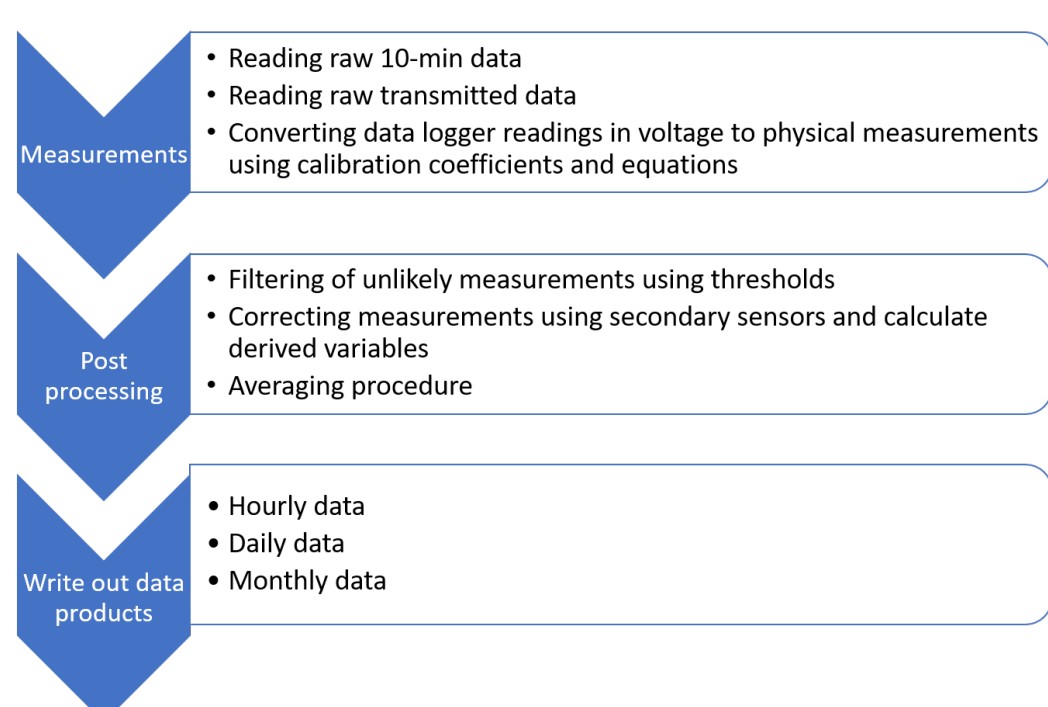

**Figure 3.** Illustration of the AWS data processing chain.

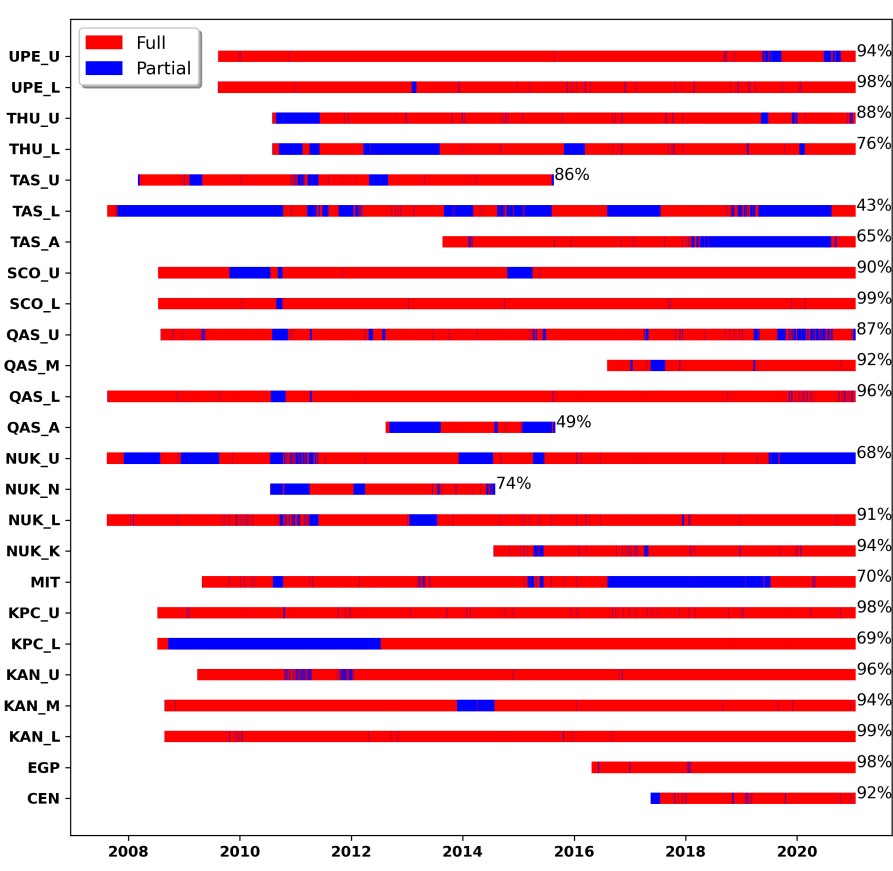

**Figure 4.** Combined availability of the 8 critical variables required for surface energy balance calculation from PROMICE daily products. See Appendix for data availability of each of the variables.

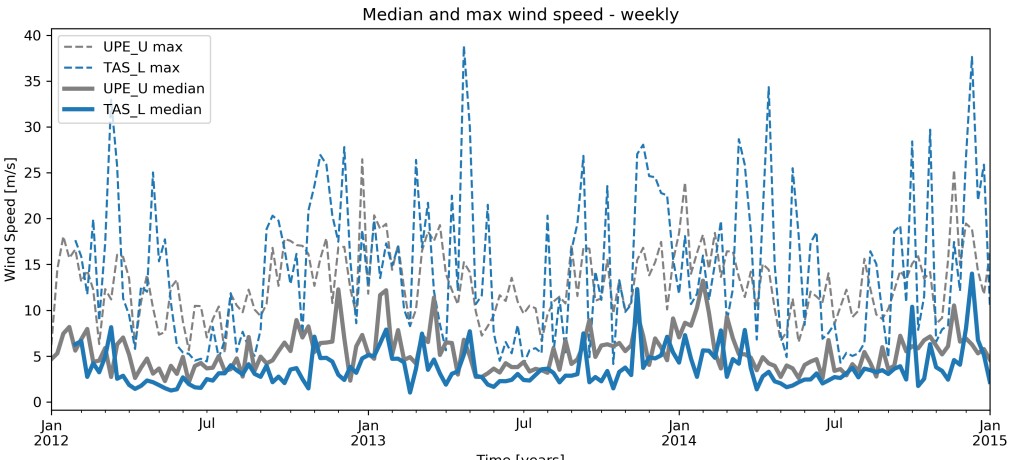

**Figure 5.** Weekly median wind speeds vs maximum 10-min wind speed within that week



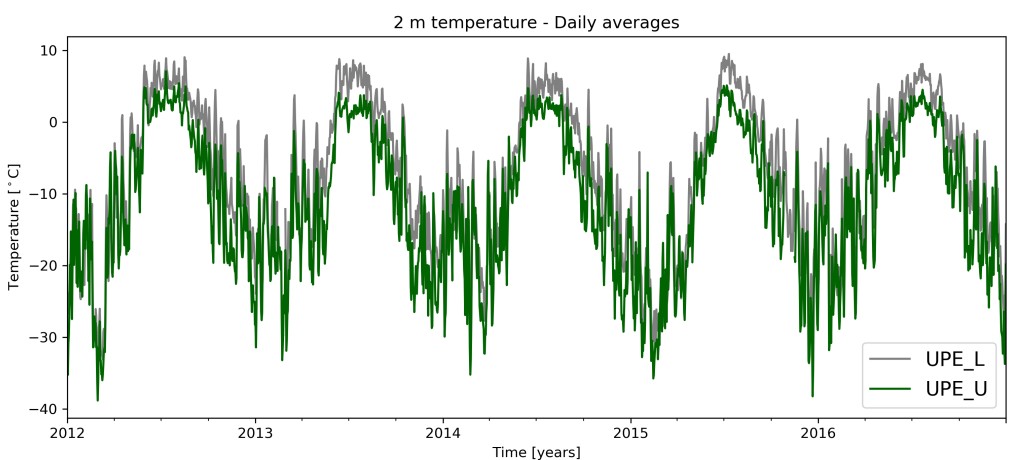

**Figure 6.** Daily air temperatures from UPE_L and UPE_U.



**Figure 7.** Estimated surface energy balance components for UPE_U.

**Table 1.** Metadata for the PROMICE automatic weather station network. Latitude, longitude and elevation are derived from automated GPS measurements in summer 2016, or during the last weeks of operation if discontinued.

| Station name | Latitude | Longitude | Elevation | Start date | Last Visit |
|---|---|---|---|---|---|
| | (°N) | (°W) | (m asl.) | (YYYY-MM-DD) | (YYYY-MM-DD) |
| KPC_L | 79.9108 | 24.0828 | 370 | 2008-07-17 | 2019 |
| KPC_U | 79.8347 | 25.1662 | 870 | 2008-07-17 | 2019 |
| EGP | 75.6247 | 35.9748 | 2660 | 2016-05-01 | 2019 |
| SCO_L | 72.2230 | 26.8182 | 460 | 2008-07-21 | 2020 |
| SCO_U | 72.3933 | 27.2333 | 970 | 2008-07-21 | 2020 |
| MIT | 65.6922 | 37.8280 | 440 | 2009-05-03 | 2019 |
| TAS_L | 65.6402 | 38.8987 | 250 | 2007-08-23 | 2020 |
| TAS_U[3] | 65.6978 | 38.8668 | 570 | 2007-08-15 | 2015-08-13 |
| TAS_A | 65.7790 | 38.8995 | 890 | 2013-08-28 | 2020 |
| QAS_L | 61.0308 | 46.8493 | 280 | 2007-08-24 | 2020 |
| QAS_M | 61.0998 | 46.8330 | 630 | 2016-08-11 | 2020 |
| QAS_U | 61.1753 | 46.8195 | 900 | 2008-08-07 | 2020 |
| QAS_A[3] | 61.2430 | 46.7328 | 1000 | 2012-08-20 | 2015-08-24 |
| NUK_L | 64.4822 | 49.5358 | 530 | 2007-08-20 | 2020 |
| NUK_U | 64.5108 | 49.2692 | 1120 | 2007-08-20 | 2020 |
| NUK_K[1] | 64.1623 | 51.3587 | 710 | 2014-07-28 | 2020 |
| NUK_N[3] | 64.9452 | 49.8850 | 920 | 2010-07-25 | 2014-07-25 |
| KAN_B[2] | 67.1252 | 50.1832 | 350 | 2011-04-13 | 2020 |
| KAN_L | 67.0955 | 49.9513 | 670 | 2008-09-01 | 2020 |
| KAN_M | 67.0670 | 48.8355 | 1270 | 2008-09-02 | 2020 |
| KAN_U | 67.0003 | 47.0253 | 1840 | 2009-04-04 | 2020 |
| UPE_L | 72.8932 | 54.2955 | 220 | 2009-08-17 | 2020 |
| UPE_U | 72.8878 | 53.5783 | 940 | 2009-08-17 | 2020 |
| THU_L | 76.3998 | 68.2665 | 570 | 2010-08-09 | 2019 |
| THU_U | 76.4197 | 68.1463 | 760 | 2010-08-09 | 2019 |
| THU_U2 | 76.3903 | 68.1101 | 744 | 2017-05-22 | 2019 |

On peripheral glacier[1] On land[2] Discontinued[3]



**Table 2.** Threshold values used in the filtering process for each measured variable.

| Instrument type | Manufaturer | Model | Accuracy (Unit) | Maintenance schedule |
|---|---|---|---|---|
| Barometer | Campbell Scientific | CS100 / Setra 278 | ±2.0 (hPa) | 5 years |
| Thermometer, aspirated | Rotronic in Rotronic assembly | MP100H-4-1-03-00-10DIN | ±0.1 (K) | 5 years |
| Hygro-/thermometer, aspirated | Rotronic in Rotronic assembly | HygroClip HC2 or HC2-S3 | ±0.1 (K) ±0.8% (rh) | visit |
| Anemometer | R.M. Young | 05103-5 | ±0.2 (ms-1) or 1 (%) of reading | 3 years |
| Radiometer | Kipp & Zonen | CNR1 or CNR4 | ±10 (%) | 3 years |
| Sonic ranger (2) | Campbell Scientific | SR50A | ±1 (cm) or ±0.4 (%) of reading | 1-2 years |
| Pressure transducer | Ørum & Jensen in GEUS assembly | NT1400 or NT1700 | ±2.5 (cm) | 5 years |
| Thermistor string | GEUS | RS PRO Termistor, 100 kΩ | ±0.9 (%) | 5 years |
| Inclinometer | HL Planar in GEUS assembly | NS-25/E2 | 0.6 (%) | 5 years |
| GPS antenna | Trimble/Tallysman | SAF5270-G/TW4020 | 2.5 (m) | 5 years |
| Iridium modem | NAL Research | 9602-LP | - | 5 years |
| Iridium antenna | Campbell Scientific | 30741 | - | 5 years |
| Batteries (4x28 Ah) | Panasonic | LC-XC1228P | - | 5 years |
| Solar panel | RS PRO | RS PRO 10 W | - | 5 years |



**Table 3.** PROMICE AWS displacement statistics from monthly average GPS data

| Station name | First valid date (YYYY-MM-DD) | Latest valid date (YYYY-MM-DD) | Time span (yr) | Displacement (m) | Displacement rate (m yr$^{-1}$) | Elevation change (m) |
|---|---|---|---|---|---|---|
| KPC_L | 2008-11-15 | 2020-06-15 | 11.6 | 80 | 6.9 | -15 |
| KPC_U | 2008-08-15 | 2020-06-15 | 11.8 | 170 | 14.3 | -2 |
| EGP | 2016-07-15 | 2020-06-15 | 3.9 | 150 | 38.2 | -1 |
| SCO_L | 2008-08-15 | 2017-07-15 | 8.9 | 749 | 84.1 | -14 |
| SCO_U | 2008-08-15 | 2012-01-15 | 3.4 | 386 | 112.9 | -8 |
| MIT | 2009-05-15 | 2020-09-15 | 11.3 | 581 | 51.2 | -31 |
| TAS_L | 2008-11-15 | 2020-07-15 | 11.7 | 198 | 17.0 | -29 |
| TAS_U | 2008-11-15 | 2015-07-15 | 6.7 | 340 | 51.0 | 3 |
| TAS_A | 2015-09-15 | 2018-09-15 | 3.0 | 275 | 91.5 | -6 |
| QAS_L | 2009-09-15 | 2020-06-15 | 10.7 | 120 | 11.1 | -55 |
| QAS_U | 2008-08-15 | 2020-04-15 | 11.7 | 622 | 53.3 | -21 |
| QAS_M | 2016-09-15 | 2020-08-15 | 3.9 | 129 | 32.9 | -12 |
| QAS_A | 2013-09-15 | 2015-02-15 | 1.4 | 121 | 85.4 | -1 |
| NUK_L | 2007-11-15 | 2020-07-15 | 12.7 | 1104 | 87.2 | -69 |
| NUK_U | 2008-11-15 | 2020-08-15 | 11.7 | 1508 | 128.4 | -21 |
| NUK_N | 2010-11-15 | 2014-07-15 | 3.7 | 84 | 22.9 | -4 |
| NUK_K | 2015-08-15 | 2020-07-15 | 4.9 | 2 | 0.4 | -8 |
| KAN_L | 2008-09-15 | 2020-08-15 | 11.9 | 1267 | 106.3 | -33 |
| KAN_M | 2008-09-15 | 2020-08-15 | 11.9 | 1240 | 104.1 | -1 |
| KAN_U | 2009-04-15 | 2020-08-15 | 11.3 | 597 | 52.7 | -11 |
| UPE_L | 2009-09-15 | 2020-07-15 | 10.8 | 17 | 1.6 | -20 |
| UPE_U | 2009-09-15 | 2020-07-15 | 10.8 | 2197 | 202.8 | -62 |
| THU_L | 2014-10-15 | 2020-06-15 | 5.7 | 26 | 4.6 | -6 |
| THU_U | 2016-08-15 | 2020-06-15 | 3.8 | 24 | 6.3 | -2 |



**Table 4.** Threshold values used in the filtering process for each measured variable.

| Variable | Units | Low threshold | High threshold |
|---|---|---|---|
| Pressure | hPa | 650 | 1100 |
| All temperatures | °C | -80 | 30 |
| Relative humidity | % | 0 | 100 |
| Wind speed | m s$^{-1}$ | 0 | 100 |
| Wind direction | ° | 0 | 360 |
| Downward shortwave radiation | W m$^{-2}$ | -10 | 1500 |
| Upward shortwave radiation | W m$^{-2}$ | -10 | 1000 |
| Downward longwave radiation | W m$^{-2}$ | 50 | 500 |
| Upward longwave radiation | W m$^{-2}$ | 50 | 500 |
| Sensor boom height | m | 0.3 | 3.0 |
| Stake assembly height | m | 0.3 | 8.0 |
| Pressure transducer assembly | m | 0 | 30 |
| Boom tilt in both directions | ° | -30 | 30 |
| Latitude | °N | 60 | 83 |
| Longitude | °W | 20 | 70 |
| Elevation | m | 0 | 3000 |
| Fan current | mA | 0 | 200 |
| Battery voltage | V | 0 | 30 |





Table 5: Short description of all the variables in our data products. An updated version of this short description is kept as a README.txt file in the data product download folder

| Variable in hourly (H), daily (D) and monthly (M) data products | Units | In data product | Short description |
| --- | --- | --- | --- |
| Year | - | H, D, M | - |
| MonthOfYear | - | H, D, M | Month of year during which measurements are taken and averaged. |
| DayOfMonth | - | H, D | Day of month during which measurements are taken and averaged. |
| HourOfDay(UTC) | UTC | H | Hour of day during which measurements are taken and averaged. |
| DayOfYear | - | H, D, M | Day of year during which measurements are taken and averaged. |
| DayOfCentury | - | H, D, M | Day of century during which measurements are taken and averaged. |
| AirPressure(hPa) | hPa | H, D, M | Barometric pressure in logger enclosure. |
| AirTemperature(C) | °C | H, D, M | Primary air temperature. Measurement height is approximately HeightSensorBoom – 0.1 m, or 2.6 m over bare ice surfaces. |
| AirTemperatureHygroClip(C) | °C | H, D, M | Secondary air temperature. Measurement height is approximately HeightSensorBoom – 0.1 m, or 2.6 m over bare ice surfaces. |
| RelativeHumidity(%) | % | H, D, M | Relative humidity wrt water/ice above/below freezing. Measurement height is approximately HeightSensorBoom –10 cm, or 2.6 m over bare ice surfaces. |
| SpecificHumidity(g/kg) | $g\ kg^{-1}$ | H, D, M | Calculated from RelativeHumidity. |
| WindSpeed(m/s) | $m\ s^{-1}$ | H, D, M | Measurement height is approximately HeightSensorBoom + 0.4 m, or 3.1 m over bare ice surfaces. |
| WindDirection(d) | ° | H, D, M | Measurement height is approximately HeightSensorBoom + 0.4 m, or 3.1 m over bare ice surfaces. |

**Table 5 – continued from previous page**

| Variable in hourly (H), daily (D) and monthly (M) data products | Units | In data product | Short description |
|---|---|---|---|
| SensibleHeatFlux(W/m2) | W m$^{-2}$ | H, D, M | Calculated using gradients of wind speed, and temperature between the surface and measurement level. Aerodynamic surface roughness for momentum is set to 0.001 m. |
| LatentHeatFlux(W/m2) | W m$^{-2}$ | H, D, M | Calculated using gradients of wind speed and humidity between the surface and measurement level. Aerodynamic surface roughness for momentum is set to 0.001 m. |
| ShortwaveRadiationDown(W/m2) | W m$^{-2}$ | H, D, M | Measurement height is approximately HeightSensorBoom + 0.1 m, or 2.8 m over bare ice surfaces. |
| ShortwaveRadiationDown_Cor(W/m2) | W m$^{-2}$ | H, D, M | Tilt-corrected values calculated from ShortwaveRadiationDown. |
| ShortwaveRadiationUp(W/m2) | W m$^{-2}$ | H, D, M | Measurement height is approximately HeightSensorBoom + 0.1 m, or 2.8 m over bare ice surfaces. |
| ShortwaveRadiationUp_Cor(W/m2) | W m$^{-2}$ | H, D, M | Tilt-corrected values calculated from ShortwaveRadiationUp. |
| Albedo_theta<70d | - | H, D, M | Surface albedo calculated from ShortwaveRadiationDown_Cor and ShortwaveRadiationUp_Cor using values obtained for solar zenith angles below 70°. |
| LongwaveRadiationDown(W/m2) | W m$^{-2}$ | H, D, M | Measurement height is approximately HeightSensorBoom + 0.1 m, or 2.8 m over bare ice surfaces. |
| LongwaveRadiationUp(W/m2) | W m$^{-2}$ | H, D, M | Measurement height is approximately HeightSensorBoom + 0.1 m, or 2.8 m over bare ice surfaces. |
| CloudCover | % | H, D | Estimated from LongwaveRadiationDown and AirTemperature. |
| SurfaceTemperature(C) | °C | H, D | Calculated from LongwaveRadiationUp and LongwaveRadiationDown. Surface longwave emissivity is set to 0.97. |

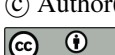



**Table 5 – continued from previous page**

| Variable in hourly (H), daily (D) and monthly (M) data products | Units | In data product | Short description |
|---|---|---|---|
| HeightSensorBoom(m) | m | H, D, M | Measured at approximately 0.1 m below the sensor boom. The sensitivity of sonic ranger readings to air temperature is removed. |
| HeightStakes(m) | m | H, D | Measured on a boom connecting aluminum stakes drilled into ice/firn. The sensitivity of sonic ranger readings to air temperature is removed. |
| DepthPressureTransducer(m) | m | H, D | Typically drilled >10 m into ice, reduces as ablation occurs. |
| DepthPressureTransducer_Cor(m) | m | H, D | Air pressure contributions eliminated from DepthPressureTransducer. |
| AblationPressureTransducer(mm) | mm | D | Daily ablation estimate from pressure transducer. Only in the daily file. |
| IceTemperature1-8(C) | °C | H, D | Subsurface temperature installed at 1, 2, 3, 4, 5, 6, 7 and 10 m depth at ablation-area sites. Note that the thermistor strings in the ablation area will melt out. |
| TiltToEast(d) | ° | H, D | Station tilt towards the east. Station may have rotated. |
| TiltToNorth(d) | ° | H, D | Station tilt towards the north. Station may have rotated. |
| TimeGPS(hhmmssUTC) GPS time stamp. | UTC | H, D | |
| LatitudeGPS(degN) | °N | H, D, M | Daily and monthly averages are calculated only using HorDilOfPrecGPS values smaller than 1. |
| LongitudeGPS(degW) | °W | H, D, M | Daily and monthly averages are calculated only using HorDilOfPrecGPS values smaller than 1. |
| ElevationGPS(m) | m | H, D, M | Daily and monthly averages are calculated only using HorDilOfPrecGPS values smaller than 1. |
| HorDilOfPrecGPS | - | H, D | GPS horizontal dilution of precision (HDOP) value. |



**Table 5 – continued from previous page**

| Variable in hourly (H), daily (D) and monthly (M) data products | Units | In data product | Short description |
|---|---|---|---|
| LoggerTemperature(C) | °C | H, D | Temperature measured by the data logger in the enclosure at 1-1.5 m above the bare ice surface. |
| FanCurrent(mA) | mA | H, D | Current drawn for ventilation of the temperature and humidity assembly. Normal values exceed 100 mA. |
| FanOK(%) | % | M | Percentage of time with sufficient ventilation of the temperature and humidity assembly. Only in the monthly file. |
| BatteryVoltage(V) | V | H, D | Voltage of the four 28 Ah batteries. Ventilation of the temperature and humidity assembly, and GPS positioning, stop below 11.5 V. |



## Appendix A: Sensor tables

### A1  Instrument information, accuracy, and power consumption

#### A1.1  Barometer

**Table A1.** Barometer: details from the manufacturer for Setra CS100 Barometric pressure sensor (Model 278).

| Parameter | Value | Unit |
|---|---|---|
| Measurement Range | 600 – 1100 | mb |
| Operating Temperature Range | –40 to 60 (–40 to 140) | °C (°F) |
| Storage Temperature Range | –60 to 120 (–76 to 248) | °C (°F) |
| Proof Pressure | 1500 | mb |
| Burst Pressure | 2000 | mb |
| Humidity Range non-condensing | (up to 95%) | RH |
| Media Compatibility non-corrosive, non-condensing air or gas Resolution: | 0.01 | mb |
| Total Accuracy: | ±0.5 @ 20 °C | mb |
| | ±1.0 @ 0 to 40 °C | |
| | ±1.5 @ –20 to 50 °C | |
| | ±2.0 @ –40 to 60 °C | |
| Linearity | ±0.4 | mb |
| Hysteresis | ±0.05 | mb |
| Repeatability | ±0.03 | mb |
| Long-term Stability: | ±0.1 per year | mb |



**Thermometer and Hygrometer**

Rotronic MP102H with Pt100 (±0.1 K) and HC2-S3 (or HC2) probe (±0.1 K, ± 0.8% rh, at 23°C ± 5 K), housed in a RS12T aspirated shield. Accuracy and factory measurement error of Rotronic probes: The Rotronic system uses ventilated weather and

**Table A2.** Thermometer and Hygrometer: details from the manufacturer Rotronics.

| Probe type | Thermometer | Hygrometer | |
| --- | --- | --- | --- |
| Pt100 | ±0.1 K | - | |
| HC2-S3 | ±0.1 K | ± 0.8% rh | at 23°C ± 5 K |
| HC2 | ±0.1 K | ± 0.8% rh | at 23°C ± 5 K |

radiation shields RS12T with a 12 VDC fan. Due to the white housing of the radiation shield the influence of thermal radiation
5   on the measurements of temperature and humidity is reduced to a minimum. The shield also offers optimum protection in
stormy weather, even against horizontally driven rain and snow. The fan is supplied by a separate cable.



**Anemometer**

**Table A3.** Anemometer: details from the manufacturer Young, model 05103.

| | Wind Speed | |
| --- | --- | --- |
| **Parameter** | **Value** | **Unit** |
| Range | 0-50 (0-112) m/s (mph) | |
| Accuracy | ±0.2 (±0.4) or 1% of reading | m/s (mph) |
| Starting threshold | 0.4 (0.9) | m/s (mph) |
| Distance constant | 2.1 (6.9), 63% recovery | m (ft) |
| Output | ac voltage (three pulses per revolution) 90 Hz (1800 rpm) = 9.2 m/s (20.6 mph) | |
| Resolution | (0.1024 m s-1) / (scan rate in seconds) | |
| | Wind Direction | |
| **Parameter** | **Value** | **Unit** |
| Mechanical range | 0-360 | ° |
| Electrical range | 355 (5 open) | ° |
| Accuracy | ±3 | ° |
| Starting threshold | 0.5 (1.0) at 10° displacement | m/s (mph) |
| Distance constant | 1.2 (3.9), 50% recovery | m (ft) |
| Damping ratio | 0.45 | |
| Damped Natural Wavelength | 4.9 (16.1) | m (ft) |
| Undamped Natural Wavelength | 4.4 (14.4) | m (ft) |
| Output | Analog dc voltage from potentiometer (resistance 10 kOhm). Linearity is 0.25%. Life expectancy is 50 million revolutions. | |
| Voltage | Power switched excitation voltage supplied by datalogger. | |





**Radiometer**

Kipp & Zonen CNR1 and CNR4. CNR4 is a four-component net radiometer for accurate and reliable measurements. There are four separate signal outputs and the integrated temperature sensors can be used to calculate the net radiation. The CNR4 combines two pyranometers for solar radiation with two pyrgeometers for infrared measurements. The upper pyrgeometer has a silicon meniscus dome so that water rolls off and the field of view is 180 °. The design is lightweight and the white sun shield reduces solar heating of the instrument body. Although similar to CNR4, the older CNR1 has a slightly different instrument body and measurement range (see Tables below), but perform with similar accuracy. We do not flag the products with respect to which instrument type we used for that each station setup. We therefore assume the same accuracy for both CNR1 and CNR4. Kipp & Zonen's CM3 ISO-class, thermopile pyranometer, CG3 pyrgeometer, PT100 RTD

**Table A4.** Thermometer and Hygrometer: details from the manufacturer Rotronics.

| Parameter | Value | Value | Unit |
|---|---|---|---|
| Sensors | CNR 1 | CNR 4 | |
| Pyranometer Spectral Response | 305 to 2800 | 305 to 2800 | nm |
| Pyrgeometer Spectral Response | 5000 to 50,000 | 4500 to 42,000 | nm |
| Response Time | 18 | < 18 | S |
| Temperature Dependence of Sensitivity | - | < 4 (-10° to +40°C) | % |
| Sensitivity Range | 7 to 15 | 5 to 20 | $\mu$ V W$^{-1}$ m$^2$ |
| Pyranometer Output Range | 0 to 25 | 0 to 15 | mV |
| Pyrgeometer Output Range | ±5 | ±5 | mV |
| Expected Accuracy for Daily Totals | ±10 | ±10 | % |
| Non-Linearity | - | < 1 | % |





**Thermistor**

Table A5. Thermistor: details from the manufacturer. Fabricated at GEUS, the thermistor strings based on resistors.

| Parameter | Value | Unit |
|---|---|---|
| Maximum Operating Temperature | +150 | °C |
| Minimum Operating Temperature | -80 | °C |
| Resistance @ 25 °C | 100 | kω |
| Temperature Coefficient Type | NTC | |
| Thermal Time Constant | 10 | S |
| Tolerance | ±0.9 | % |





**Inclinometer**

**Table A6.** Inclinomter: details from the manufacturer HL planar Technik model NS-25/E2.

| Parameter | Value | Unit |
|---|---|---|
| Measuring range | ± 25 | ° |
| Measuring axes | Two (x/y) orthogonal orientated | |
| Resolution | 0.002 | ° |
| Precision | 0.6 | % |
| Banking sensitivity | < 1.5 | % |
| Temperature stability: | | |
| Zero point | 0.002 | K |
| Sensitivity | 0.005 | K |





**Sonic rangers**

The accuracy of the SR50A sonic ranger given by the manufacturer (Campbell Scientific) is: ± 1 cm or ± 0.4% of the measuring height after temperature correction.

**Pressure transducer**

5  The PROMICE AWSs are equipped with an Ørum & Jensen NT1400/NT1700 pressure transducer assembly that allow us to monitor ice surface height change due to ablation. The pressure transducer sensor has an accuracy of 2.5 cm given by the manufacturer (Ørum & Jensen Elektronik A/S).

**GPS**

Single frequency GPS is equipped. Built into Iridium 9602-LP modem. Receiver Type: NEO-6Q, 1575.42 MHz (L1), 16-
10  channel, C/A code Accuracy: 2.5 m CEP Update Rate: 5 Hz Start-up Times: 1 second hot-starts, 28 seconds warm- and cold-starts Sensitivity: -160 dBm