# Peer review of "PROMICE automatic weather station data"

_Earth System Science Data, 2021_

## Author Response (AR1)

Dear David Carlson,

We would like to thank you, one anonymous reviewer, and Jacob C. Yde for thoughtful and useful comments on our manuscript.

In the following, we provide a point-by-point response to all the reviewer comments. The reviewer comments are in black while our replies are in green for clarity. We have also included a tracked changed version of the manuscript.

Best regards,

Robert Fausto on behalf of all co-authors.

Review of

PROMICE automatic weather station data

by Fausto and others, submitted to ESSD

General

This paper presents an overview of the near-surface climate, surface mass balance, and GPS data collected by the PROMICE network of AWS on the Greenland ice sheet. The PROMICE data have been transformational for the study of the climate of the Greenland ice sheet, the evaluation of (regional) climate models, and the validation of satellite data. The paper is generally well and concisely written, and the figures/tables are clear. So I recommend publication after some minor issues have been addressed, see below.

Major comments

p. 1, l 10, section tilt correction: It would be illustrative to provide the average corrections that were obtained for SWin for the various AWS.

Answer: We have included a table (New Table 3) with average bias and the standard deviation of that bias. We have added the following text in section 3.3.2 under tilt correction: "Table 3 illustrates the average bias or correction made for the incoming solar radiation based on Equation (11). The standard deviation indicates for most AWSs that the

average correction is relatively minor (below 15 Wm⁻²), while few AWSs have corrections values spread out over a wider range (Table 3)."

Section 4: instead of presenting time series of two selected AWS, which comes across as somewhat arbitrary, I would have preferred a table with climatological averages, including the SEB terms, simply to demonstrate that the stations are suitable to base a climatology on in the first place and that they are in climatologically distinct regions, which is the main asset of PROMICE. This could be accompanied by a brief analysis of basic climate information e.g. near-surface lapse rate, SWin/SW_TOA, etc.

Answer: We recognize that a table with climatological averages, including the SEB terms, is useful for demonstrating and quantifying the climatologically distinct regions of the AWS network. However, to include an actual and proper data analysis is, in our opinion, beyond the scope of this dataset description paper, so we will refrain from doing that. Also, even a brief analysis of basic climate information will quickly make this already long dataset description paper even longer. We just want to give examples on how the dataset can be used, so we would like to keep that part. We have included a table in the appendix (Table B1) and a subsection, called "Data climatology: Average and standard deviation", to quantify and describe the averages and their variation. We include the meteorological parameters, and the corrected and estimated surface energy balance terms.

Minor and textual comments

p1, l. 18: "Presently, the PROMICE AWS data are not included in any reanalysis product such as ERA5, aiding studies with an independent assessment of the performance of regional climate models, and other numerical models that aim 20 to quantify surface mass or energy fluxes (Fettweis et al., 2020)." Has this been a deliberate choice? Is it an option to reverse this in the future, to improve the reanalysis products themselves?

Answer: It has not been a deliberate choice, but circumstances and certain challenges made it problematic, so we chose to wait. It is our goal to make the AWS data available through WMO in the near future.

p. 2, l. 11: "allows full coverage of the ice sheet," suggest: "allows coverage in various climate zones" as full coverage is evidently not possible and/or aimed for.

Answer: Done

p. 3, l. 23: "snow and ice ablation/accumulation" suggest: "snow ablation/accumulation and ice ablation"

Answer: Done

p. 8, l. 11: "to quantify ice dynamics" suggest "to quantify ice floe velocity

Answer: Done

Review 2

The authors present a well-structured technical description of the PROMICE AWS variables and instrumentation. The presentations of the data processing chain, the measured variables and each instrument are all robust, and it is easy for readers to find relevant information in the text and tables. I did not notice any technical or scientific issues that need to be addressed. However, the text could be improved as shown by my list of linguistic comments below.

1,3: Change "is" to "are"

Answer: Done

1,4: Change "to" to "of"

Answer: Done

1,13: Change "in" to "by"

Answer: Done

1,14-15: Too many brackets. This is a general issue throughout the manuscript

Answer: Done

1,17-18: This statement should be supported by one or more references

Answer: Done

2,17-18: Be consistent with respect to the usage of capitalized letters in "Greenland Ice Sheet" or "Greenland ice sheet". Please check the entire manuscript for consistency

Answer: We will use "Greenland ice sheet"

2,25: Insert space before "The"

Answer: Done

3,16-17: This sentence regarding maintenance visits seems to be misplaced as this subchapter concerns the design of the tripod. Consider moving this sentence to the last paragraph in section 2.2

Answer: Done

4,6: Change to "encodes"

Answer: Done

4,17: Typo in "production"

Answer: Fixed

5,5: It would be more informative for the readers if the name of the barometer manufacturer is mentioned in the body text

Answer: No, the reader is referred to Table 2 and the appendix for more information.

7,13: Insert "Figure" before "2"

Answer: Done

7,17: Insert "of" in "… in-/outflow of antifreeze …"

Answer: Done

7,18: Insert "Figure" before "2"

Answer: Done

8,21: Insert a hyphen in "Post-processing"

Answer: Done

9,13: Typo in "respectively"

Answer: Done

9,18: Be consistent throughout the manuscript in whether "Van" should start with a lowercase or uppercase letter

Answer: Ok, but we are using the Copernicus template (copernicus.cls) when citing others work.

9,18: Change ";" to "and"

Answer: Again, we use the Copernicus template (copernicus.cls) when citing others work.

9,27: Insert a comma after "m" and change "are" to "is"

Answer: Done

9,28-29: Rephrase this sentence to make it understandable

Answer: Done. Rephrased into: "We use the stability correction functions $\psi_{u,T,q}$ from Equation 12 in Holtslag and De Bruin (1988) for stable atmospheric conditions, while we follow Paulson (1970) for unstable conditions."

10,12: Use past tense here – change to "emphasized"

Answer: Done

10,13: Who are "they"? Rephrase this sentence

Answer: Done. Rephrased into: "Fausto et al. (2016a, b) investigated the use of an unrealistically high $z_0$ to get agreement between SEB closure and observed ablation rates during extreme sensible and latent heat-driven melt events"

10,18-19: Change to "For a horizontal radiation sensor, the direct beam …"

Answer: Done

11,15; Change "are" to "is"

Answer: Done

11,16: Insert "." after "AWS"

Answer: Done

11,19: Typo in "manufacturer" and change to "estimates"

Answer: Done

11,22: Be consistent in the use of "Fig" or "Figure"

Answer: Done. We use "Figure"

11,28: Insert "the" before "field"

Answer: Done

12,13: Change to "Eighteen out of 26 stations …"

Answer: Done

12,27: Use lowercase first letter in "piteraq"

Answer: Done

13,17: It would be good with more consistency in the comma usage. For instance, the authors use comma after "Also, …" (13,14) and "Below, …" (13,15), but no comma after "Here …" (13,17)

Answer: We agree. There should be comma after "Here" in (13,17). We have checked the manuscript again for consistency.

15,13: Move "e.g." to the bracket

Answer: Done

15,18: Insert "of" before "26"

Answer: Done

Table 1: Could you be more specific with respect to the MM or DD of the last visits?

Answer: Done

Table 2: The caption text is wrong

Answer: Correct caption now.

Table A1: Reconsider the use of uppercase letters in column 1 - this comment is also relevant for other tables. I would also recommend left-aligned text in column 1. Use lowercase first letter in "barometric" in the caption

Answer: We would like to stick with uppercase letters in column 1 throughout the tables.

Table A2: Use lowercase first letter in "hygrometer" in the caption

Answer: Done

36,4: Insert a comma after "shield"

Answer: Done

41,2: Delete ":"

Answer: Done

41,5: Use more formal language instead of "... that allow us to ..."

Answer: Ok

41:9-11: Rewrite this text into full sentences.

Answer: Done